# Prevalence and spectrum of germline *BRCA1* and *BRCA2* mutations in multiethnic cohort of breast cancer patients in Brunei Darussalam

Siti Nur Idayu Matusin ¤a, Nuramalina Mumin, Hazirah Zainal Abidin ¤b,
Fatin Nurizzati Mohd Jaya ¤c, Zen Huat Lu, Mas Rina Wati Haji Abdul Hamid *

Pengiran Anak Puteri Rashidah Sa'adatul Bolkiah (PAPRSB) Institute of Health Sciences, Universiti Brunei Darussalam, Gadong, Negara Brunei Darussalam

¤a Current address: Halalan Thayyiban Research Centre, Universiti Islam Sultan Sharif Ali, Tutong, Negara Brunei Darussalam
¤b Current address: Ministry of Health, Commonwealth Drive, Berakas, Negara Brunei Darussalam
¤c Current address: ICC, AstraZeneca, Gaithersburg, Maryland, United States of America
* rina.hamid@ubd.edu.bn

## Abstract

This is the first genetic study of its kind in Brunei Darussalam. *BRCA1* and *BRCA2* genes are the most well-known and well described predictors of hereditary breast cancer due to their clinical importance. This study aimed to identify the prevalence and mutation spectrum of the *BRCA1* and *BRCA2* germline mutations among 120 unselected series of Brunei breast cancer patients. We screened the entire coding region of *BRCA1* and *BRCA2* gene using Sanger sequencing and next-generation sequencing methods and identified three pathogenic and one likely pathogenic mutations in the *BRCA2* gene. Of the 120 patients, 6 (5%) were *BRCA2* carriers which confirm that *BRCA2* carriers are more common in the Asian population compared to the Caucasian population. One *BRCA2* mutation observed only in the Chinese ethnicity of the Brunei breast cancer population suggest a probability of the mutation being a founder effect in the Southern Chinese population. Brunei *BRCA2* carriers were more likely to have a positive family history of breast and/or ovarian cancers and have more than one family members in the first-degree relatives diagnosed with breast cancer.

## Introduction

Breast cancer is the most common cancer affecting women worldwide with varying incidence rates across the world [1]. Among the breast cancer risk factors, genetic predisposition confers the highest risk in breast cancer progression [2]. About 5–10% of breast cancers are inherited with approximately 30% of the inherited breast cancers were attributed to germline mutations in high penetrance breast cancer susceptibility genes, Breast Cancer susceptibility genes type 1 (*BRCA1*) [3] and Breast

**Data availability statement:** All relevant data are within the manuscript and its Supporting Information files.

**Funding:** -MRWHAH - [UBD/PNC2/2/RG/1(186)] - Universiti Brunei Darussalam - ubd.edu.bn - The funders had no role in study design, data collection and analysis, decision to publish, or preparation of the manuscript.

**Competing interests:** The authors have declared that no competing interests exist.

Cancer susceptibility gene type 2 (*BRCA2*) [4]. *BRCA1* and *BRCA2* are tumour suppressor genes (TSGs) that function by suppressing the growth of tumour cells via multiple DNA damage and repair pathways in the cells [5–10]. Notably, the BRCA1 protein is multi-functional as it also regulates cell cycle through a number of mechanisms [5,6,8,10–15].

*BRCA1* and *BRCA2* germline mutation carriers had an increased cumulative risk of 72% and 69%, respectively to develop breast cancer by 80 years old [16]. *BRCA1* and *BRCA2* carriers would also have an increased risk of developing ovarian, prostate, and pancreatic cancers [16]. Breast cancer patients who inherit germline mutations in these genes, in particular the *BRCA1* carriers are commonly associated with triple negative breast cancer (TNBC), diagnosed at early onset (≤40 years), diagnosed with bilateral breast cancer, having family history of breast and/or ovarian cancers in the first- and second-degree relatives, and having an ovarian cancer [17–21]. The discovery of these two genes being linked to breast and ovarian cancers has led to the increased importance of genetic testing where continuous research and development had ultimately resulted in *BRCA1* and *BRCA2* genetic tests being available at a more affordable cost and with a higher sensitivity and specificity [22]. Moreover, increased awareness on available personalised treatment for affected patients such as olaparib, an oral poly ADP ribose polymerase (PARP) inhibitor, which has been shown to provide a significant benefit over standard therapy among the *BRCA* mutation-carrier patients [23,24], has led to the increasing demand for rapid *BRCA* testing preferably at the first diagnosis [22]. This rapid increase caused pressure for diagnostic laboratories to provide a genetic test with a shorter turnaround time [22]. The two most commonly used platform for DNA sequencing are Sanger sequencing and Next-generation sequencing (NGS). Although NGS has largely overtaken Sanger sequencing due to its cost-effective ability in screening a larger set of samples in parallel and simultaneous screening of multiple cancer susceptibility genes in one sample, Sanger sequencing is still used in laboratories today when the main objective is to screen a single gene only.

Brunei Darussalam is a small country (5,765 sq km) on the Borneo Island, located in the South-east Asia (SEA) region, bordering the South China Sea and East Malaysia [25]. The population is estimated over 400,000, comprises 66% Malay, 10% Chinese, 3% other indigenous and 21% other ethnic groups [25]. Breast cancer was one of the top three leading causes of cancer mortality among women in Brunei Darussalam with an incidence rate (age-standardized rate [ASR]) of 55.9 per 100,000 women which is the second highest among countries in the SEA region, but considerably lower compared to Western Europe (ASR, 90.7) [1,26]. Notably, the incidence rate of breast cancer in Brunei varies by ethnicity, with the highest rate observed in Chinese (ASR, 60.4) [27], followed by Malays (ASR, 48.1) [27], and Others (ASR, 12.3) [28]. There has not been any formal study conducted on finding the contribution of genetic and non-genetic factors in the rising incidence of breast cancer in Brunei. Therefore, studying various aspects of breast cancer in Brunei patients could help to understand and determine the probable cause that leads to the rise in the incidence of the disease in the population, and in planning a

better health care for the Brunei population. The contribution of genetics in breast cancer, specifically the involvement of susceptibility genes, has been continuously researched in the Western and developed Asian countries, resulting in many reports on the spectrum of variants within these susceptibility genes from all over the world. In the SEA region, the mutation spectrum of *BRCA1* and *BRCA2* genes in breast cancer has been studied in Singaporean [18,19,29–31], Malaysian [21,32–37], Filipino [38,39], Vietnamese [40,41], Thai [42,43], and Indonesian [44,45] populations leading to the discovery of novel pathogenic variants from different ethnicities, suggesting that mutation spectrum of the *BRCA1* and *BRCA2* genes are still understudied in the region. Furthermore, a number of the variants identified in the SEA breast cancer populations has also been found in other populations, such as the African, European, and American suggesting that they are not specific to the SEA populations, highlighting the complexity of population genetics and the potential for variants to shift in frequency due to factors like migration, social influences, and historical events. These findings underscore the importance of conducting haplotype analysis to better understand the true origin of recurrent variants. Interestingly, most of the variants identified in breast cancer population in East Malaysia, which is also located on the Borneo Island [33] were not identified in that of the West Malaysian population, further suggesting genetic differences between the East and West Malaysian populations.

Some of the variants identified in the Filipino breast cancer population who mostly were a combination of Malay and other population ancestry have also been proposed to be founder mutations due to the shared haplotype markers [39]. Currently, there are no genetic mutation data that report on the frequency of any genes related to breast cancer in Bruneian patients with or without family history. Thus, genetic contribution in breast cancer and other related cancers in Bruneian population remains scarcely-investigated. In this study, we investigated a population-based unselected series of Brunei breast cancer patients to determine the prevalence and spectrum of *BRCA1* and *BRCA2* mutations using Sanger and next-generation sequencing methods and assess the association of pathogenic variants only with sociodemographic, clinicopathological, and family history characteristics of the population.

## Materials and methods

### Study population

The study population includes unselected incident and prevalent breast cancer patients seen from 18th May 2012 until 30th January 2013, and from 23rd May 2016 until 19th November 2018 at The Brunei Cancer Centre (TBCC), Jerudong Park Medical Centre (JPMC), the only cancer referral hospital in Negara Brunei Darussalam. Terminally ill patients were excluded from the study due to confounding variables and ethical concerns. Among approximately 567 patients attending clinics during this period of time, a total of 164 patients were approached where 121 patients consented as the final study participants. All study participants signed informed consent document. Peripheral blood samples and demographic and family history data were collected from the consenting patients. One patient was later excluded due to insufficient DNA sample to complete the *BRCA1* and *BRCA2* mutation analysis. Finally, 120 patients were included for the final analysis. Retrospective review of the study participants' medical and histopathology records was conducted at TBCC with authorised permission from relevant authorities from 1st November 2018 to 9th January 2019. The records were available in the form of hard printed copy. The study was approved by the Brunei Darussalam Ministry of Health's Medical and Health Research &Ethics Committee (MHREC).

### *BRCA1* and *BRCA2* mutational screening

S1 Fig shows a flow chart of the strategy used to detect mutations in the *BRCA1* and *BRCA2* genes in the study population. The samples from the first 66 study participants (henceforth referred to as Batch 1) were analysed by Sanger sequencing while the samples from the last 54 study participants (henceforth referred to as Batch 2) were sequenced by next-generation sequencing (NGS).

**Sanger sequencing.** Samples sequenced using Sanger sequencing utilised two workflows, PCR amplified with designed primers, and PCR amplified using EasySeq™ PCR Plates for *BRCA1/2* Sequencing (Nimagen, The Netherlands).

In the first workflow, the gDNA extracted using Wizard Genomic DNA purification kit (Promega) was amplified using AmpliTaq® 360 DNA Polymerase kit (Applied Biosystems) prior to sequencing. The details of the primers and PCR conditions used in this study have been described elsewhere [46–48]. The PCR products were resolved onto 1% agarose gel for evaluation of successful amplifications without contamination. The remaining PCR products were then purified using QIAGEN QIAquick® PCR Purification Kit.

In the second workflow, the gDNA extracted was amplified using the EasySeq™ PCR Plates for *BRCA1/2* Sequencing (Nimagen, The Netherlands) following the manufacturer's protocol. The plate was used for sequencing to screen for mutations in the complete coding region including ±50 bp up- and downstream of each *BRCA1* and *BRCA2* genes' coding exon. All the primers in the columns were tailed with universal tails – forward primers with -21M13 (5'-TGTAAAACGAC GGCCAGT-3') and reverse primers with M13Rev (5'-CAGGAAACAGCTATGACC-3'). PCR products from columns 1–10 were purified via ethanol/sodium acetate precipitation, and the pellets were resuspended with sterile water.

The purified PCR products from both workflows were used as the template for the Sanger sequencing reactions following the BigDye Terminator® v3.1 Cycle Sequencing Kit protocol (Applied Biosystems, USA). The sequencing products were purified using the Axygen AxyPrep™ Mag Dye Clean Up Kit. *BRCA1* and *BRCA2* Sanger sequencing was performed using the ABI 3500 Genetic Analyzer (Applied Biosystems, USA). The sequence data collected were processed and analysed using the 3500 Sequencing Analysis Software v5.2. Purified PCR products from the first workflow were sent for Sanger sequencing (First BASE Laboratories, Malaysia).

**Next-generation sequencing (NGS).** The DNA samples sequenced by NGS were outsourced to Cancer Research Malaysia (CRM) for *BRCA1, BRCA2, TP53,* and *PALB2* targeted panel sequencing. The HBOC_4_v2 gene panel used was developed by CRM and the University of Melbourne and was used to screen for coding exons ±2 bp intronic sequence of the *BRCA1* (NM_007294.3)*, BRCA2* (NM_000059.3)*, PALB2* (NM_024675.3), and *TP53* (NM_000546.4) genes. Bioinformatic analysis was performed by CRM and the variant results excluding neutral polymorphisms were delivered upon completion.

**Bioinformatics analysis.** Sanger sequencing data obtained using Sequencing Analysis Software v5.2 and from First BASE Laboratories were aligned to the reference sequences of the gene obtained from the NCBI GenBank (*BRCA1* NCBI RefSeq = NG_005905.2 and *BRCA2* NCBI RefSeq = NG_012772.1) and nucleotide database (*BRCA1* NCBI RefSeq = NM_007294.3 and *BRCA2* NCBI RefSeq = NM_000059.3) using Variant Reporter Software v1.0 (Applied Biosystems, USA). All variants identified were annotated using the same software. Variants called were filtered based on quality scores (QUAL) and depth of coverage. Specifically, variants with a read depth of less than 20x or a QUAL score below 30 were excluded from further analysis to minimise false positives.

For samples sequenced by NGS, although CRM have analysed and validated the results, the data were re-analysed as neutral polymorphisms were not included in the provided results. The attainment of these results ensures the generation of a complete genetic variation profile for the patients. Integrative Genomics Viewer (IGV) v2.5.2 (Broad Institute) was used to view the binary files provided by CRM. The data were analysed using the public server at usegalaxy.eu [49]. Prior to variant analysis, statistics for BAM dataset (Galaxy Version 2.0.2 + galaxy1) was first generated using samtools stats to assess the base alignment quality of the data. The default parameters were applied except for the set distribution coverage where the maximum coverage was set to 100000, and the coverage step was set to 10. The quality of the samples and data were considered good if the PHRED quality score is ≥ 20 and MQ0 is ≥ 20. Variants were called using: [1] varscan (Galaxy Version 2.4.2) applying the default parameters except that the minimum read depth was set to 10 and the minimum supporting reads set to 4, and [2] bcftools call (Galaxy Version 1.9 + galaxy 1) applying the default parameters except that the output variant sites were set to yes and the output was set to uncompressed VCF. The data generated by bcftools

call were filtered using VCFfilter. All variants identified from NGS were annotated using the SNPeff Eff tool (Galaxy Version 4.3 + T.galaxy1). Acceptable variants must be present in (a) both members of read-pairs, (b) DP4 ≥ 15% of read-pairs.

All variants identified from Sanger sequencing and NGS were annotated according to the nomenclature used by the Human Genome Variation Society (HGVS) recommendation guidelines, using the A of the ATG translation initiation codon as nucleotide +1. All identified missense variants were analysed *in silico* using SIFT, PolyPhen-2, CADD, FATHMM-MKL, and DANN to predict the effect of amino acid substitution. Each prediction tool scored the missense variant as damaging or benign/neutral. All variants identified in this study were also checked against the NCBI ClinVar, Varsome, and population frequency databases (gnomAD and 1000 genome).

All variants identified in this study, underwent thorough assessment and review of available evidence (e.g., database information, and *in silico* predictions) following the American College of Medical Genetics and Genomics and the Association for Molecular Pathology (ACMG) standards and guidelines for the interpretation of sequence variants [50] to arrive at a final variant classification of either pathogenic, likely pathogenic, variant of uncertain significance (VUS), likely benign, and benign. In this study, a variant was considered a pathogenic damaging mutation if it was a protein-truncating mutation caused by deleterious or frameshift mutation, or a missense mutation which has a confirmed association with the disease, or a missense variant which has been classified as likely pathogenic according to the ACMG standards and guidelines. All pathogenic mutations identified using Sanger sequencing were confirmed by repeating the Sanger sequencing using an independent sample, while pathogenic mutations identified using NGS were re-sequenced using Sanger sequencing. We selectively Sanger sequenced only the pathogenic mutations identified using NGS via our own ABI 3500 Genetic Analyzer. Notably, CRM had confirmed all variants other than neutral polymorphisms by Sanger sequencing.

### Multiplex ligation-dependent probe amplification (MLPA)

Due to resource constraints, only samples from Batch 1 were screened for large genomic rearrangements by MPLA using the SALSA MLPA P002 BRCA1 and SALSA MLPA P090 BRCA2 probe mix following the manufacturers' protocol (MRC-Holland, Netherland). This limitation was necessary to ensure the feasibility of completing the analyses within the available time frame and fund. The MLPA analyses were performed by DNA Fragment Analysis on the ABI 3500 Genetic Analyzer. Data obtained were comparatively analysed using Coffalyser Net software v.140721.1958 (MRC-Holland, Netherland). Notably, future studies with resources could aim to expand this analysis to additional batches.

### Statistical analyses

Continuous data were presented as mean ± standard deviation (SD) or median and range where applicable. Mann-Whitney test was used to compare differences between the medians in two or more independent groups respectively. Chi-square or Fisher exact test was used to analyse the association between two independent variables in the population. The statistical analysis was performed using SPSS version 17.0 software for Windows. A p value <0.05 was considered statistically significant.

## Results

This was the first study of its kind in the Brunei breast cancer patients. Of the 120 recruited breast cancer patients, 94 (78.3%) were Malays, 19 (15.8%) were Chinese, and 7 (5.8%) were others. The mean age at diagnosis was 49.1 ± 10.51 years, with peak of breast cancer incidence occurred in the 51–60 years old age group, and one (0.8%) was a male breast cancer (Table 1). Among the 120 patients, two [2] were related, consisting of a mother and daughter pair, and two [2] others were step sisters with the same father. The sisters did not carry pathogenic mutations.

While the majority of patients had no personal history of cancer other than the currently diagnosed cancer, seven (5.8%), patients had personal history of bilateral breast cancers, two (1.7%) had ovarian cancer, one (0.8%) had endometrium cancer, and one (0.8%) had liver cancer. Family history of breast and/or ovarian cancers was assessed based on whether the affected family members were first and/or second-degree relatives. Of the 120 cases, 36 (30.0%) patients

**Table 1. Demographic characteristics, personal and family history of cancer of study population (*n* = 120)f.**

| Characteristics | Mean (SD) | Frequency (%) |
|---|---|---|
| **Age at diagnosis (year)** | 49.1 (10.51) | |
| **Gender** | | |
| Male | | 1 (0.8) |
| Female | | 119 (99.2) |
| **Ethnicity** | | |
| Malay | | 94 (78.3) |
| Chinese | | 19 (15.8) |
| Others | | 7 (5.8) |
| **Age at first diagnosis (year)** | | |
| ≤ 30 | | 2 (1.7) |
| 31-40 | | 28 (23.3) |
| 41-50 | | 36 (30.0) |
| 51-60 | | 40 (33.3) |
| ≥ 61 | | 14 (11.7) |
| **Cancer history** | | |
| **Personal history of cancer** | | |
| Bilateral breast cancer | | 7 (5.8) |
| Ovarian cancer | | 2 (1.7) |
| Endometrium cancer | | 1 (0.8) |
| Liver cancer | | 1 (0.8) |
| **Family history of cancer** | | |
| Breast and/or ovarian cancer in 1° and 2° relatives | | 36 (30.0) |
| Breast and/or ovarian cancer in 3° and 4° relatives | | 9 (7.5) |
| Other cancer in any degree of relatives | | 28 (23.3) |
| No family history of cancer at all | | 47 (39.2) |
| **No of 1° relatives with breast cancer** | | |
| 0 | | 100 (83.3) |
| 1 | | 17 (14.2) |
| ≥ 2 | | 3 (2.5) |
| **No of 2° relatives with breast cancer** | | |
| 0 | | 104 (86.7) |
| 1 | | 14 (11.7) |
| ≥ 2 | | 2 (1.7) |
| **No of 1° relatives with ovarian cancer** | | |
| 0 | | 115 (95.8) |
| ≥ 1 | | 5 (4.2) |
| **No of 2° relatives with ovarian cancer** | | |
| 0 | | 118 (98.3) |
| ≥ 1 | | 2 (1.7) |

1°, First-degree; 2°, Second-degree; 3°, Third-degree; 4°, Fourth-degree; SD, Standard Deviation.

had family history of breast and/or ovarian cancers in their first and/or second-degree relatives, while 47 (39.2%) had no family history of cancers at all. Of the patients with family history of breast cancer, 17 (14.2%) had one first-degree family member affected with breast cancer, and 3 (2.5%) had two first-degree relatives affected with breast cancer. For patients

with family history of ovarian cancer, 5 (4.2%) had only one affected member in the first-degree relative, and 2 (1.7%) had one affected second-degree relatives (Table 1).

### BRCA1 and BRCA2 mutations

Of the 95 variants identified in the study population, 29 (30.5%) and 66 (69.5%) were *BRCA1* and *BRCA2* variants respectively (S1–S4 Tables). Of the 29 *BRCA1* and 66 *BRCA2* variants, 11 (37.9%) and 22 (33.3%) were novel respectively (S2–S4 Tables). No large rearrangements were detected in this study. All variants reported in this study that were identified using NGS were subsequently validated by Sanger sequencing in-house or outsourced. The results from both platforms demonstrated 100% concordance, confirming the accuracy of variant detection. NGS is highly sensitive and allows for the simultaneous screening of multiple genes, making it particularly effective for detecting rare and novel variants. However, it may be prone to sequencing artifacts, which is why Sanger sequencing is commonly used for validation. In this study, our findings reaffirmed that both methodologies were robust, with no discrepancies observed between NGS and Sanger sequencing results.

**Prevalence of pathogenic mutations and VUS.** Our study identified three fraeshift *BRCA2* deleterious mutations in five [5] patients. These frameshift mutations led to the introduction of premature stop codons that resulted in truncated and non-functional BRCA2 protein (S1 Table, Figs 1 and S2). Additionally, we identified one likely pathogenic *BRCA2* missense mutation in one [1] patient (S2 Table and Fig 1). Therefore, the overall prevalence of germline *BRCA1* and *BRCA2* carriers among Brunei breast cancer patients were 0% and 5%, respectively. All identified pathogenic and likely pathogenic mutations have been previously reported and were rare in the general population (<0.05% in gnomAD and 1000 Genome). Notably, one of the *BRCA2* deleterious mutations (c.5164_5165delAG) appeared to be a recurrent mutation.

Excluding one patient with simultaneous germline pathogenic mutations, our study also identified 9 *BRCA1* and 32 *BRCA2* VUS in 10 and 31 patients, respectively (S3 Table). Thus, the prevalence of *BRCA1* and *BRCA2* VUS carriers among Brunei breast cancer patients was 8.3% and 25.8%, respectively. Among these, four [4] *BRCA1* VUS and 14 *BRCA2* VUS were novel, while the remaining variants have been previously reported. Almost all reported VUS lacked population data, and those with available data were found to be rare in the general population (<0.05% in gnomAD and 1000 Genome) (S3 Table).

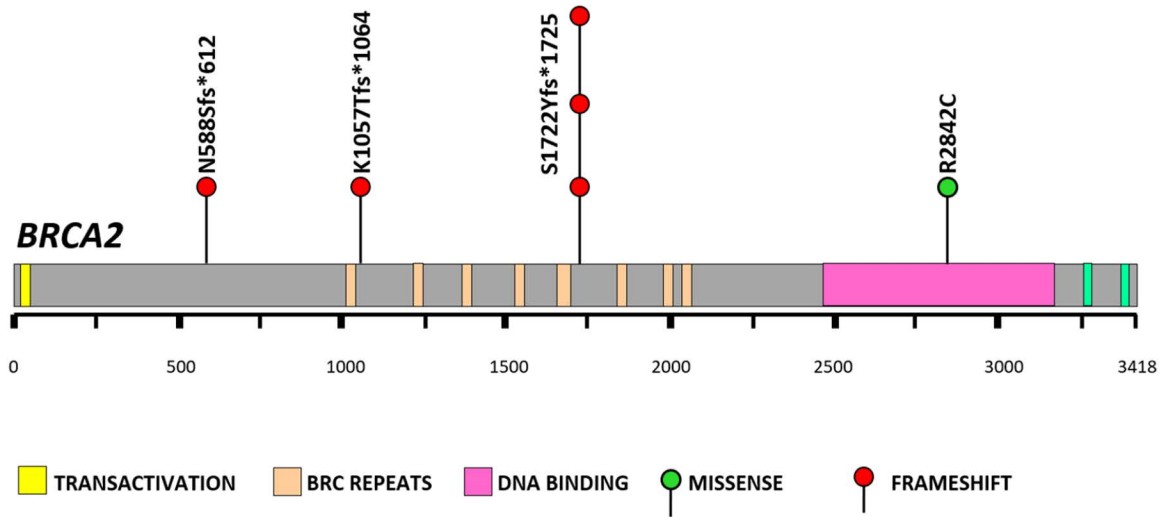

**Fig 1. Schematic presentation of identified *BRCA2* pathogenic mutations among Brunei breast cancer cases.**

**Characteristics of Brunei's germline pathogenic *BRCA* carriers.** S5 Table shows the characteristics of the Brunei breast cancer patients with deleterious and damaging missense mutations. Collectively, one (16.7%) of the carriers was male while the rest were female. Two (33.3%) of the affected were diagnosed with bilateral breast cancer. The molecular subtype of the first breast cancer diagnosis showed that one (16.7%) patient was TNBC, one (16.7%) was Luminal A, while the rest were Luminal B. Two (33.3%) of the carriers had an age at onset of ≤40 years while the others were ≥40 years, and one (16.7%) of the carriers had no family history of cancers at all. The proportion of our *BRCA1/2* carriers by ethnicity was 3/19 (15.8%) and 3/94 (3.2%) in Chinese and Malay, respectively.

There is no significant difference observed between the overall median age of diagnosis among carriers (48 years) and non-carriers (49 years) ($p = 0.609$, Table 2). Brunei *BRCA2* carriers were found to be more likely to have family history of breast and/or ovarian cancer and have ≥1 affected family members in the first-degree with breast cancer ($p = 0.027$ and $p = 0.001$ respectively, Table 2). There are no significant associations between Brunei *BRCA2* carriers and other selected clinical characteristics in Table 2.

## Discussion

Prior to introducing a cancer genetic testing in a community, genetic variation profile of the said community must first be known to have an overview of the contribution of genetics in the population. Although cancer incidence and mortality in Brunei Darussalam have been increasing for the past 10 years, there are still no data available on the proportion of Brunei cancer patients carrying germline or somatic mutations in their cancer predisposing genes. *BRCA1* and *BRCA2* gene frequencies in breast and ovarian cancer patients in other East-Asian populations have been continuously researched on which had led to the discovery of potential founder mutations [29,39,51,52]. Notably, identifying who should be offered a cancer genetic testing in the Asian community remains challenging as the currently available guidelines for testing have been shown to omit at least 20% of patients carrying *BRCA* mutations from being tested [21], suggesting the testing criteria on *BRCA1* and *BRCA2* germline mutations which have been well established in the Caucasian population were not readily applicable to the Asian population.

Our study, which is the first genetic study of its kind in Brunei, showed that the overall combined frequency of germline pathogenic *BRCA1* and *BRCA2* mutation carriers among 120 recruited Brunei's breast cancer patients was 5% ($n = 6$), which is lower than those reported at 12.3% in women of European descendent [52]. Our data are relatively similar to previously reported prevalence from other Asian countries in particular from the South-East Asian region which is 4.7% in Malaysia [21] and 5.1% in the Philippines [39] except for Singapore [19] and Vietnam [40] which were reported at 17.4% and 0.8%, respectively. Interestingly, other Asian countries outside of the South-East Asian region had reported a relatively higher prevalence of *BRCA1* and *BRCA2* mutations which was almost similar (or higher for some) to the findings reported in European women with 14.0% in Korea [53] and 24.7% in Pakistan [54]. Japan [55], Hong Kong [51,56] and China [20] however, reported a prevalence of 4.2%, 7.9% to 8.8%, and 8.3%, respectively. Comparatively, all these countries had reported prevalence of germline mutations in both *BRCA1* and *BRCA2* genes while all germline *BRCA* mutations identified in our study were attributed to *BRCA2* gene only. Our findings were consistent with those from all the aforementioned Asian countries, where most of the germline pathogenic *BRCA* mutations in the Brunei breast cancer patients were identified in the *BRCA2* gene compared to *BRCA1* which was reported to be more affected in the European women [57].

Our study showed that one recurrent deleterious frameshift mutation, *BRCA2*c.5164_5165delAG (S1722Yfs*1725), was identified in 3 (2.5%) patients of Chinese ethnicity. In particular two of the affected patients, 4G-004 and 4G-003, were related to each other (mother and daughter, respectively). This mutation was suggested to have a founder effect in the Southern China province population [56]. While it was unknown whether our Chinese index cases ancestry were originally from the Southern China, there was still a probability that the occurrence of this mutation might be linked to a common ancestral origin. Moreover, this mutation has also been reported in the Chinese ethnicity in Malaysia [34] and Singapore [18] which suggests the probability of the founder mutation theory. However, it is acknowledged that the

**Table 2. Association analysis of Brunei breast cancer patients with pathogenic mutation with selected clinical characteristics.**

| | Total (*n*=120) | | BRCA2 carriers (*n*=6) | | Non-BRCA2 carriers (*n*=114) | | *p*-value [a] [b] |
|---|---|---|---|---|---|---|---|
| | *n* | % | *n* | % | *n* | % | |
| **Age at diagnosis (years)** | | | Median (range) 48 (31–59) | | Median (range) 49 (27–71) | | 0.609 |
| **Age range (years)** | | | | | | | |
| ≤40 | 30 | 25.0 | 2 | 33.3 | 28 | 24.6 | 0.741 |
| 41–50 | 36 | 30.0 | 1 | 16.7 | 35 | 30.7 | |
| 51–60 | 40 | 33.3 | 3 | 50.0 | 37 | 32.5 | |
| >60 | 14 | 11.7 | 0 | 0 | 14 | 12.3 | |
| **Ethnicity** | | | | | | | |
| Malay | 94 | 78.3 | 3 | 50.0 | 91 | 79.8 | 0.089 |
| Chinese | 19 | 15.8 | 3 | 50.0 | 16 | 14.0 | |
| Others | 7 | 5.8 | 0 | 0 | 7 | 6.1 | |
| **Bilateral breast cancer** | | | | | | | |
| Yes | 8 | 6.7 | 2 | 33.3 | 6 | 5.3 | 0.051 |
| No | 112 | 93.3 | 4 | 66.7 | 108 | 94.7 | |
| **Estrogen Receptor (ER)** | | | | | | | |
| Positive | 72 | 60.0 | 4 | 66.7 | 68 | 59.6 | 1 |
| Negative | 48 | 40.0 | 2 | 33.3 | 46 | 40.4 | |
| **Progesterone Receptor (PR)** | | | | | | | |
| Positive | 62 | 51.7 | 4 | 66.7 | 58 | 50.9 | 0.681 |
| Negative | 58 | 48.3 | 2 | 33.3 | 56 | 49.1 | |
| **Her2** | | | | | | | |
| Positive | 73 | 60.8 | 3 | 50.0 | 70 | 61.4 | 0.678 |
| Negative | 47 | 39.2 | 3 | 50.0 | 44 | 38.6 | |
| **Triple negative breast cancer (TNBC)** | | | | | | | |
| Yes | 15 | 12.5 | 1 | 16.7 | 14 | 12.3 | 0.559 |
| No | 105 | 87.5 | 5 | 83.3 | 100 | 87.7 | |
| **Family history of cancer [c]** | | | | | | | |
| Yes | 72 | 60.0 | 5 | 83.3 | 67 | 58.8 | 0.400 |
| No | 48 | 40.0 | 1 | 16.7 | 47 | 41.2 | |
| **Family history of breast and/or ovarian cancer** | | | | | | | |
| Yes | 45 | 37.5 | 5 | 83.3 | 40 | 35.1 | **0.027** |
| No | 75 | 62.5 | 1 | 16.7 | 74 | 64.9 | |
| **Number of affected first-degree relatives with breast cancer** | | | | | | | |
| 0 | 100 | 83.3 | 2 | 33.3 | 98 | 86.0 | **0.001** |
| 1 | 17 | 14.2 | 2 | 33.3 | 15 | 13.2 | |
| ≥2 | 3 | 2.5 | 2 | 33.3 | 1 | 0.9 | |
| **Number of affected second-degree relatives with breast cancer** | | | | | | | |
| 0 | 104 | 86.7 | 5 | 83.3 | 99 | 86.8 | 0.585 |
| 1 | 14 | 11.7 | 1 | 16.7 | 13 | 11.4 | |
| ≥2 | 2 | 1.7 | 0 | 0 | 2 | 1.8 | |
| **Number of affected first-degree relatives with ovarian cancer** | | | | | | | |
| 0 | 115 | 95.8 | 5 | 83.3 | 110 | 96.5 | 0.230 |
| ≥1 | 5 | 4.2 | 1 | 16.7 | 4 | 3.5 | |

*(Continued)*

**Table 2.** (Continued)

| | Total (*n* = 120) | | *BRCA2* carriers (*n* = 6) | | Non-*BRCA2* carriers (*n* = 114) | | *p*-value [a] [b] |
|---|---|---|---|---|---|---|---|
| | *n* | % | *n* | % | *n* | % | |
| **Number of affected second-degree relatives with ovarian cancer** | | | | | | | |
| 0 | 118 | 98.3 | 5 | 83.3 | 113 | 99.1 | 0.098 |
| ≥1 | 2 | 1.7 | 1 | 16.7 | 1 | 0.9 | |
| **Vital status** | | | | | | | |
| Alive | 95 | 79.2 | 4 | 66.7 | 91 | 79.8 | 0.603 |
| Deceased | 25 | 20.8 | 2 | 33.3 | 23 | 20.2 | |

[a]Mann-Whitney test or Fisher's Exact test whichever appropriate.

[b]Statistically significant *p*-values are indicated in bold.

[c]Inclusive of breast, ovarian and other cancers.

probability could be slightly low due to the fact that two out of three of the carriers of this mutation were relatives where the variant most likely would have been inherited from the maternal side of the family.

Although the other two deleterious frameshift *BRCA2* mutations identified in this study, and each only occured in one (0.8%) of the study population, the mutations c.3170_3174delAGAAA (K1057Tfs*1064) and c.1763_1766delATAAA (N588Sfs*612) had each been reported as a founder mutation in the French-Canadian [58] and Colombian [59] populations respectively. However, definitive conclusions regarding the origin of these mutations in our study population would require haplotype analysis, which was not conducted in this study. Given that the carriers in our study were of Malay ethnicity and their family histories did not indicate a common ancestral origin with the previously-mentioned populations, it is possible that these mutations arose independently or were introduced through historical genetic events.

Interestingly, in the South-East Asia (SEA) population, the *BRCA2* mutation c.1763_1766delATAA (N588Sfs*612) was only ever reported in a breast cancer patient in Sarawak, Malaysia who was diagnosed at <40 years old with triple negative breast cancer and family history of breast cancer [33]. This observation suggests that affected populations have a different population genetic predisposition towards breast cancer. A pathogenic mutation determined as a founder effect in one population may only be detected in a small frequency in other populations of different ancestral origin. Further genetic and haplotype studies would be necessary to determine whether K1057Tfs*1064 and N588Sfs*612 mutations represent independent occurrences or are linked to broader historical genetic patterns. For the two patients identified with these mutations, there is a 50% chance that their affected family members carried the same *BRCA2* mutation that could lead to the development of breast cancer.

The three previously-mentioned frameshift mutations (S1 Table) introduced premature stop codons, which ultimately result in truncated BRCA proteins with loss of function. The mutations lead to the destabilisation of *BRCA* mRNA through nonsense-mediated decay, causing haplo-insufficiency and allelic imbalance [60]. This in turn lead to a significant reduction in the expression ratio between the mutant and the wild-type alleles [60], resulting in decreased levels of both *BRCA2* transcripts and proteins. The allelic imbalance contributes to an increased risk of breast cancer [60]. The truncated BRCA2 proteins exhibited loss of critical functional domains: [1] the BRC repeat domains – completely absent in c.1763_1766delATAAA, and spanning BRC-2 to BRC-8 in c.3170_3174delAGAAA and BRC-6 to BRC-8 in c.5164_5165delAG; [2] the DNA binding domain (DBD); and [3] the two nuclear localisation signals (NLS) in the C-terminal region (S2 Fig). The eight highly conserved BRC repeat domains and the DBD in BRCA2 are essential for mediating interactions with single-stranded and double-stranded DNA, facilitating the loading of RAD51 onto DNA during homologous recombination (HR) repair pathway. In this pathway, BRCA2 localises to DNA double-strand breaks through complexes with other proteins, including BRCA1 and PALB2, before mediating RAD51 loading onto single-stranded DNA

to initiate repair. Loss of these RAD51-interacting domains impairs DNA repair via the error-free HR pathway, allowing the error-prone non-homologous end joining (NHEJ) pathway to predominate. Deletion of all RAD51-interacting domains in mice has been shown to cause embryonic lethality, however deletion of several BRC repeat motifs resulted in a less severe phenotype, with affected mice succumbing to cancer at a very young age [61]. The DBD in BRCA2 also plays a role in transcriptional regulation by binding single-stranded DNA and recruiting histone modifier, thereby initiating transcription [10]. Loss of the two NLS in the C-terminal region due to the truncation results in the mutant BRCA2 being retained in the cytoplasm, as it cannot be translocated into the nucleus, rendering it non-functional [62].

A likely pathogenic missense mutation, c.8524C>T (R2842C) was identified in a patient with no family history of cancer (S2 and S5 Tables). This variant was located in the BRCA2 DBD, suggesting a potential impact on protein function. In a study, where this variant was classified as VUS a Homology-Directed Repair (HDR) functional assay in the BRCA2 DBD was conducted. The latter exhibited 95% to 99% probabilities of pathogenicity, and despite this, R2842C was classified as intermediate pathogenic as none of the known pathogenic standards went below 99% [63]. Additionally, in another study using a mouse embryonic stem cell (mESC)-based assay the intermediate pathogenicity of R2842C was further tested. It could complement the loss of cell viability while its capacity for HDR was reduced by more than 50% compared to wild-type BRCA2-expressing cells [64]. The assay also correctly discriminated between pathogenic (class 4/5) and non-pathogenic (class 1/2) variants that were previously classified using genetic and clinical data. Interestingly, another study identified R2842C as a homozygous hypomorphic variant in primary ovarian insufficiency (POI) without cancer nor Fanconi Anaemia (FA) pathologies in the patient or the family [65]. From functional HR experiments, the ACMG-AMP classification and the Sherloc scoring framework adopted in the study, the variant was considered pathogenic [65]. Findings from these three studies show that evidence from functional assays are essential to determine the level of pathogenicity of variant R2842C in causing breast cancer and the described POI. Family history and *in silico* classification could not be rendered sufficient to associate pathogenicity of the variant with the disease. In our study, the variant R2842C was identified in patient 2G-022 with unilateral breast cancer, and without family history of cancers, POI or FA (S5 Table). ACMG classification categorised it as likely pathogenic (S2 Table). The variant most likely increased the breast cancer risk in the patient, its occurence and mechanism of risk however, warrants further functional investigations. Due to limitations in the available facilities, we were unable to conduct functional studies to further characterise the hypomorphic nature of this variant. Notably, none of the founder mutations prevalent in the Southeast Asian (SEA) population was detected in our study cohort.

Similar to other genetic variation analysis conducted in other populations, 41 (43.2%) of the identified variants in the study population were classified as VUS (S3 Table). The overall combined frequency of patients with *BRCA1* and *BRCA2* VUS in the study was 35 (29.2%), in which 6 patients have both *BRCA1* and *BRCA2* VUS. Our findings were higher than those reported in other populations, such as 5.7% in European [57], and 1.6% in Malaysian [21] women, respectively. It has been shown that the prevalence of VUS in a study population was inversely correlated with the total number of individuals tested [57]. Hence, if our total study cohort population was higher than 120, our prevalence could be relatively similar with other countries' findings as demonstrated in three other Asian countries' studies conducted comprising over 2500 breast cancer patients; namely China [20], Malaysia [21], and Japan [55].

Collectively, Brunei *BRCA2* carriers exhibited almost the same phenotypic association pattern as other *BRCA2* carriers in China [20] and Malaysia [21]. Brunei *BRCA2* carriers were found to be more likely to have family history of breast and/or ovarian cancers, and having more first-degree relatives affected with breast cancer. However, in this study their pathogenicity status was not validated via functional experiments. Consistent with other studies, our findings showed that TNBC is not significantly associated with *BRCA2* carriers [17,19,21].

It is acknowledged that this study is limited by the small size of the study population even though it was the first and largest cohort study for genetic testing in Brunei. Future study on a larger cohort of study population could confirm our findings on the prevalence of germline mutations in the Brunei breast cancer population. Moreover, it could also provide a more comprehensive data for phenotypic characteristics of predisposing carriers in our breast cancer population attributed

to each gene. Our current study showed that none of our study population tested for *BRCA1* and *BRCA2* large rearrangements showed positive for the test. Therefore, there are no data on the contribution of large rearrangement of *BRCA1* and *BRCA2* in the Brunei breast cancer population presented in this report. Given the constraint in getting resources for performing large rearrangements analysis in some of the study population (Batch 2), there is an approximately 10% probability that some carriers may have been missed in this study [19].

## Conclusions

The findings from this study have highlighted the contribution of genetics, specifically *BRCA1* and *BRCA2* genes in 120 unselected series of Brunei breast cancer population. The prevalence of germline *BRCA2*, and seemingly lack of *BRCA1,* mutations (5%) among Brunei breast cancer patients is similar to that of other Asian populations further highlighting the difference in *BRCA* genetics and associated breast cancer risk in Caucasian and Asian populations. Three [3] frameshift mutations, c.3170_3174delAGAAA (K1057Tfs*1064), c.1763_1766delATAAA (N588Sfs*612) and c.5164_5165delAG (S1722Yfs*1725) were found and confirmed as pathogenic by ACMG criteria. One [1] missense mutation, c.8524C>T (R2842C) was inferred as likely pathogenic by the ACMG criteria. The possibility of a founder effect on the concerned variants mentioned requires haplotype analysis. While the variants described were identified based on population frequency databases such as gnomAD and ClinVar, further investigation is necessary to determine whether these mutations have been reported in other studies involving other Southeast Asian populations. Comparative studies across neighbouring populations with similar ethnic backgrounds could help contextualise the significance of these findings. Future studies should investigate the contribution of germline mutations of *BRCA1* and *BRCA2* genes in a larger case-control cohort study of Brunei breast cancer population to confirm and further investigate the contribution of these germline mutations to breast cancer risk.

## Supporting information

**S1 Fig. The study strategy to detect *BRCA1* and *BRCA2* mutations.**
(TIF)

**S2 Fig. Schematic presentation of truncated BRCA2 protein in the study population.** The three identified *BRCA2* pathogenic mutations, c.1763_1766delATAAA, c.3170_3174delAGAAA and c.5164_5165delAG led to BRCA2 protein truncation at amino acids positions 612, 1064, and 1725 respectively. The truncation caused the loss of important domains affecting the function of BRCA2 predominantly in the HR-regulated DNA repair pathway. NLS; Nuclear localization signals.
(TIF)

**S1 Table. Deleterious mutations identified in the Brunei breast cancer patients.** *, Termination; AA, amino acid; AF, Allele Frequency; Freq, Frequency; FS, Frameshift; Nov, Novel; Ref, Reference; Rep, Reported.
(XLSX)

**S2 Table. Damaging missense variants identified in the Brunei breast cancer patients.** AA, amino acid; AF, Allele Frequency; B, Benign; D, Damaging; Freq, Frequency; N, Neutral; NA, Not Available; Nov, Novel; PrD, Probably Damaging; Ref, Reference; Rep, Reported; T, Tolerated.
(XLSX)

**S3 Table. Frequency and classification of missense variants identified in the Brunei breast cancer patients.** AA, amino acid; AF, Allele Frequency; B, Benign; D, Damaging; Freq, Frequency; N, Neutral; NA, Not Available; Nov, Novel; PoD, Possibly Damaging; PrD, Probably Damaging; Ref, Reference; Rep, Reported; T, Tolerated; VUS, Variant of uncertain significance.
(XLSX)

**S4 Table. Synonymous variants identified in the Brunei breast cancer patients.** AA, amino acid; AF, Allele Frequency; Freq, Frequency; NA, Not Available; Nov, Novel; Rep, Reported.
(XLSX)

**S5 Table. Characteristics of the Brunei breast cancer patients with deleterious and damaging missense mutations.** +, Positive; -, Negative; 'XX, year diagnosed or deceased; AA, Amino acid; Aff, Affected; Au, Aunt; Bil., Bilateral; Br, Breast; Bro, Brother; Ca, Cancer; Chi, Chinese; Cou, Cousin; Dau, Daughter; de., deceased; dx., diagnosed; DC, Ductal carcinoma; DCIS, Ductal carcinoma in situ; ER, Oestrogen receptor; Eth, Ethnicity; F, Female; Fa, Father; Gen, Gender; GrMo, Grandmother; Her2, Human Epidermal Growth Receptor-2; IDC, Invasive ductal carcinoma; ILC, Invasive lobular carcinoma; L, Left; Liv, Liver; Lu, Lung; M, Male; Mal, Malay; Mat, Maternal; Mo, Mother; O, Older; Ov, Ovarian; Pat, Paternal; PR, Progesterone receptor; R, Right; Rec, Rectal; Sis, Sister; Sync, Synchronous; Uni, Unilateral; Y, Younger; yr., years.
(XLSX)

## Acknowledgments

We sincerely acknowledge all the doctors, nurses and staff from TBCC for their cooperation and also to CRM and First Base Laboratories Sdn Bhd for their contributions to the study. We would like to thank our undergraduate students; Jacinda Lim Xin Yan and Hon Kar Yee; who have taken a small project under this study. Finally, we would like to express our gratitude to all patients who have participated in our research study.

## Author contributions

**Conceptualization:** Mas Rina Wati Haji Abdul Hamid.

**Data curation:** Siti Nur Idayu Matusin, Zen Huat Lu.

**Formal analysis:** Siti Nur Idayu Matusin.

**Funding acquisition:** Mas Rina Wati Haji Abdul Hamid.

**Investigation:** Siti Nur Idayu Matusin, Nuramalina Mumin, Hazirah Zainal Abidin, Fatin Nurizzati Mohd Jaya, Zen Huat Lu, Mas Rina Wati Haji Abdul Hamid.

**Methodology:** Siti Nur Idayu Matusin, Nuramalina Mumin, Hazirah Zainal Abidin, Fatin Nurizzati Mohd Jaya, Zen Huat Lu, Mas Rina Wati Haji Abdul Hamid.

**Project administration:** Siti Nur Idayu Matusin, Mas Rina Wati Haji Abdul Hamid.

**Resources:** Siti Nur Idayu Matusin, Nuramalina Mumin, Hazirah Zainal Abidin, Fatin Nurizzati Mohd Jaya, Zen Huat Lu, Mas Rina Wati Haji Abdul Hamid.

**Software:** Zen Huat Lu.

**Supervision:** Zen Huat Lu, Mas Rina Wati Haji Abdul Hamid.

**Validation:** Siti Nur Idayu Matusin, Zen Huat Lu.

**Visualization:** Siti Nur Idayu Matusin, Nuramalina Mumin, Hazirah Zainal Abidin, Fatin Nurizzati Mohd Jaya, Zen Huat Lu, Mas Rina Wati Haji Abdul Hamid.

**Writing – original draft:** Siti Nur Idayu Matusin.

**Writing – review & editing:** Mas Rina Wati Haji Abdul Hamid.

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
