## [Decision Letter · Decision Letter 0]

PONE-D-24-36034Prevalence and spectrum of germline BRCA1 and BRCA2 mutations in multiethnic cohort of breast cancer patients in Brunei DarussalamPLOS ONE

Dear Dr. Haji Abdul Hamid,

Thank you for submitting your manuscript to PLOS ONE. After careful consideration, we feel that it has merit but does not fully meet PLOS ONE’s publication criteria as it currently stands. Therefore, we invite you to submit a revised version of the manuscript that addresses the points raised during the review process.

We look forward to receiving your revised manuscript.

Kind regards,

Yonglan Zheng, Ph.D.

Academic Editor

PLOS ONE

We sincerely acknowledge all the doctors, nurses and staff from TBCC for their cooperation and also to CRM and First Base Laboratories Sdn Bhd for their contribution to the study. We would like to thank our undergraduate students; Jacinda Lim Xin Yan and Hon Kar Yee; who have taken a small project under this study. Finally, we would like to express our gratitude to all patients who have participated in our research study.  MRWHAH received a grant from Universiti Brunei Darussalam  to fund the initial part of this study [UBD/PNC2/2/RG/1(186))] 

-MRWHAH

- [UBD/PNC2/2/RG/1(186)]

- Universiti Brunei Darussalam

- ubd.edu.bn

Reviewers' comments:

Reviewer's Responses to Questions

**Comments to the Author**

1. Is the manuscript technically sound, and do the data support the conclusions?

Reviewer #1: Yes

Reviewer #2: Partly

2. Has the statistical analysis been performed appropriately and rigorously? 

Reviewer #1: Yes

Reviewer #2: No

3. Have the authors made all data underlying the findings in their manuscript fully available?

Reviewer #1: Yes

Reviewer #2: Yes

4. Is the manuscript presented in an intelligible fashion and written in standard English?

Reviewer #1: Yes

Reviewer #2: No

5. Review Comments to the Author

**Reviewer #1: ** This manuscript investigated the germline BRCA1/2 mutations among Brunei breast cancer patients and aimed to characterize the genetic contribution to breast cancer based on the mutational profiles. The authors used both NGS and Sanger sequencing to characterize the BRCA genes using peripheral blood samples. The variants were carefully examined and annotated and the association between genetic variation and demographical/clinical features of the patients was then examined. The manuscript was well written, and the supporting evidence and information was very well documented. Although the conclusion was based on small sample size, I believe this manuscript would be of interests to the scientific community.

Major comments:

- For the variants reported, could the author also report whether these were identified with Sanger or NGS, and comment on the sensitivity of mutation detection between the two platforms (within this study and/or in general)?

- Over 30% of novel variants seem a lot. Can the authors comment on the QC applied to the variant calling? Maybe adding the quality metrics of from the callers would help interpret the results.

- Copy number variation of BRCA has been reported. Did authors plan to look into that as well?

- Line 243: Was the conclusion that these variants have been reported based on the population frequency database only? Were they also found in other studies based on similar population in SEA?

- Line 309: citation needed

Minor comments:

- Line 235: space between “29BRCA1” and “66BRCA2”

**Reviewer #2: ** Dear Dr. Matusin and collaborators,

The manuscript titled “Prevalence and Spectrum of Germline BRCA1 and BRCA2 Mutations in a Multiethnic Cohort of Breast Cancer Patients in Brunei Darussalam” offers valuable insights into an understudied population. However, it needs substantial revisions before it can be considered for publication. I hope the following comments will be helpful.

1. To help ensure that the ideas and concepts in the manuscript are easily understood, it would be beneficial to conduct a thorough review of the English grammar. This will enhance clarity and make the content more approachable for readers.

2. Introduction:

a) The last paragraph on page 4 requires clearer conceptual explanation. It’s important to understand that finding recurrent variants of clinical significance in African or European populations, as well as in other Asian populations, does not necessarily mean that these variants aren't originally from Africa or Europe. Population structures can shift due to various factors, including social influences and significant events like war, immigration, and emigration. As a result, the frequencies of genetic variants—whether benign or pathogenic—can also change. Thus, haplotype analysis is crucial in determining the true origin of these recurrent variants.

b) In the first paragraph on page 5, the objective was to evaluate potential associations between demographic characteristics and the profiles of BRCA1 and BRCA2. However, it would be helpful to present this data more clearly to improve understanding and interpretation, as the data was not provided.

3. Materials and Methods

a) In the section “study population”, it was noted that terminally ill patients were excluded from the study. The reason for excluding these patients was not specified.

b) In the section titled “Multiplex Ligation-Dependent Probe Amplification (MLPA)” on page 9, it states that only the samples from batch 1 were subjected to MLPA analysis. However, it does not explain why samples from the other batches were not included in this screening, especially since Next-Generation Sequencing (NGS) has significant limitations in detecting large genomic rearrangements.

4. Results

a) The "Personal history of cancer" section in Table 1 currently provides details for only 10% of the studied cases. It would be helpful to gather and include information about the remaining 90% to provide a more comprehensive overview.

b) In Table 1, the row labeled "Number of 2nd relatives with breast cancer" indicates a sum greater than 100%. It may be beneficial to review this data for accuracy.

c) The first paragraph on page 12 needs a thorough review, as clarifying its content could enhance comprehension for readers.

d) To enhance clarity in the legends for Figure S2, it would be helpful to provide descriptions for the acronyms used. This addition could improve understanding for readers.

e) In the S1 Table, I recommend reviewing the ACMG criteria codes, as there appears to be an error in the application of the supporting codes. These variants have strong evidence of pathogenicity and have been reported previously.

f) In Table S2, it would be beneficial to revisit the ACMG codes used and the classification of the variants for clarity. Additionally, I would appreciate insights on the decision why not applying the BP1 code.

g) I recommend revising the title of the S3 table to enhance clarity and engagingness.

h) In Table S5, I recommend replacing the year of diagnosis or death with the patient's age. This adjustment would provide more clinically relevant information.

i) The section “Prevalence of pathogenic mutations and VUS” on page 13 requires significant revision.

5. Discussion

a) In the second paragraph on page 18, it is noted that two carriers of the BRCA2: c.5164_5165delAG are related. Providing more information about the total number of related and unrelated patients in the cohort would enhance the clarity of the discussion. This additional context could help in better understanding the study's findings.

b) The first paragraph on page 19 mentions that a Malay patient carried two founder mutations in the BRCA2 gene: one from French-Canadian origin and the other from Colombian origin. While family history provides compelling evidence, a valid conclusion regarding the potential origin of the BRCA2:c.3170_3174delAGAAA and BRCA2:c.1763_1766delATAAA mutations can only be reached through haplotype analysis.

c) I believe it would be beneficial to rework the last paragraph on page 19 and the first paragraph on page 20 for improved clarity and impact.

d) Page 21 requires rewriting and a review of the concept. The manuscript does not describe the functional analysis of the BRCA2 mutation: c.8524C>T. Relying solely on family history is insufficient.

6. Conclusion

a) I suggest revisiting the conclusions to enhance their clarity and accuracy. While it is noteworthy that only one BRCA2 founder mutation was identified in individuals of Chinese descent, this does not necessarily indicate that the mutation found in the Brunei patient is the same as this particular founder mutation. Furthermore, it’s valuable to acknowledge that other BRCA2 founder mutations from European and African populations were also observed in the Brunei population studied. To further improve the conclusions, the authors could explore conducting a haplotype analysis or offer a more straightforward interpretation of the current findings.

6. PLOS authors have the option to publish the peer review history of their article (what does this mean? ). If published, this will include your full peer review and any attached files.

**Do you want your identity to be public for this peer review?** For information about this choice, including consent withdrawal, please see our Privacy Policy .

Reviewer #1: No

Reviewer #2: No

---

## [Author Response · Author response to Decision Letter 1]

14 May 2025

Dr Yonglan Zheng

Academic Editor

PLOS ONE

13 May 2025

RE: Response to Reviewer Comments for Manuscript [PONE-D-24-36034]

Dear Dr Yonglan Zheng,

We sincerely appreciate the time and effort the reviewers have taken to evaluate our manuscript titled “Prevalence and spectrum of germline BRCA1 and BRCA2 mutations in multiethnic cohort of breast cancer patients in Brunei Darussalam.” We are grateful for their insightful comments and constructive suggestions, which have helped us improve the quality of our work. Below, we provide detailed responses to each comment and describe the revisions we have made accordingly. Please be informed that the line numbers are based on the file “Revised Manuscript with Track Changes - Matusin et al.

Reviewer #1 Comments and Responses:

Comment 1: For the variants reported, could the author also report whether these were identified with Sanger or NGS, and comment on the sensitivity of mutation detection between the two platforms (within this study and/or in general)?

Response: We acknowledge the reviewer’s concern and have addressed this by adding an additional column in the S1-S4 Table indicating the method used when the variants are detected, along with the description of the concordance between the two platforms. The revised section can be found on page 13, lines 269-275 and S1-S4 Table files.

Comment 2: Over 30% of novel variants seem a lot. Can the authors comment on the QC applied to the variant calling? Maybe adding the quality metrics of from the callers would help interpret the results.

Response: Thank you for this valuable suggestion. We have now added the quality metrics applied in our study. The updated content is reflected in Section Materials and Methods, pages 8-9, lines 184-186 and 191-202

Comment 3: Copy number variation of BRCA has been reported. Did authors plan to look into that as well?

Response: We appreciate the reviewer’s suggestion regarding copy number variation (CNV) of BRCA. At this stage, our preliminary findings do not indicate significant results that would warrant further investigation into CNV at this time as indicated in line 268. However, we acknowledge the relevance of this aspect and may consider it in future studies if further data support its inclusion.

Comment 4: Line 243: Was the conclusion that these variants have been reported based on the population frequency database only? Were they also found in other studies based on similar population in SEA?

Response: The conclusion for these variants are addressed in the Conclusion section, page 24, lines 488-492.

Comment 5: Line 309: citation needed

Response: The citation has been added (Line 352).

Comment 5: Line 235: space between “29BRCA1” and “66BRCA2”

Response: Amended (Line 267).

Reviewer #2 Comments and Responses:

Comment 1: To help ensure that the ideas and concepts in the manuscript are easily understood, it would be beneficial to conduct a thorough review of the English grammar. This will enhance clarity and make the content more approachable for readers.

Response: We appreciate the reviewer’s suggestion regarding language clarity. To enhance readability and ensure grammatical accuracy, we have thoroughly reviewed and revised the manuscript for improved clarity. Additionally, we have sought assistance from a native English speaker to help refine the language. We believe these revisions have made the manuscript more accessible to readers.

Comment 2a: The last paragraph on page 4 requires clearer conceptual explanation. It’s important to understand that finding recurrent variants of clinical significance in African or European populations, as well as in other Asian populations, does not necessarily mean that these variants aren't originally from Africa or Europe. Population structures can shift due to various factors, including social influences and significant events like war, immigration, and emigration. As a result, the frequencies of genetic variants—whether benign or pathogenic—can also change. Thus, haplotype analysis is crucial in determining the true origin of these recurrent variants.

Response: We have revised the explanation. The revision can be found on page 4, line 97-104.

Comment 2b: In the first paragraph on page 5, the objective was to evaluate potential associations between demographic characteristics and the profiles of BRCA1 and BRCA2. However, it would be helpful to present this data more clearly to improve understanding and interpretation, as the data was not provided.

Response: We have added clarification in our study objective where the associations are only evaluated for pathogenic variants only. The revision can be found on page 5, line 112.

Comment 3a: In the section “study population”, it was noted that terminally ill patients were excluded from the study. The reason for excluding these patients was not specified.

Response: We have added clarification on the exclusion of these patients (line 123).

Comment 3b: In the section titled “Multiplex Ligation-Dependent Probe Amplification (MLPA)” on page 9, it states that only the samples from batch 1 were subjected to MLPA analysis. However, it does not explain why samples from the other batches were not included in this screening, especially since Next-Generation Sequencing (NGS) has significant limitations in detecting large genomic rearrangements.

Response: We acknowledge the importance to conduct MLPA side by side with NGS in such study, we would have loved to do more MLPA, however we were constrained by funding and so we were not able to conduct this further. Clarification to address this comment is added on page 10, lines 225, 227-228, and 231.

Comment 4a: The "Personal history of cancer" section in Table 1 currently provides details for only 10% of the studied cases. It would be helpful to gather and include information about the remaining 90% to provide a more comprehensive overview.

Response: Thank you for the suggestion. We have included a clarification sentence on the remaining 90% of the patients as a starting sentence in the paragraph on page 12, line 253-254.

Comment 4b: In Table 1, the row labeled "Number of 2nd relatives with breast cancer" indicates a sum greater than 100%. It may be beneficial to review this data for accuracy.

Response: We appreciate the reviewer’s careful review of our data. We have rechecked the calculations both in SPSS and manually, confirming that the values are correct. The percentages are rounded according to SPSS conventions, which may result in a total slightly exceeding 100%. This is a common occurrence in percentage rounding and does not indicate an error in the data.

Comment 4c: The first paragraph on page 12 needs a thorough review, as clarifying its content could enhance comprehension for readers.

Response: The paragraph has been refined accordingly.

Comment 4d: To enhance clarity in the legends for Figure S2, it would be helpful to provide descriptions for the acronyms used. This addition could improve understanding for readers.

Response: We have added the acronyms accordingly in line 699.

Comment 4e: In the S1 Table, I recommend reviewing the ACMG criteria codes, as there appears to be an error in the application of the supporting codes. These variants have strong evidence of pathogenicity and have been reported previously.

Comment 4f: In Table S2, it would be beneficial to revisit the ACMG codes used and the classification of the variants for clarity. Additionally, I would appreciate insights on the decision why not applying the BP1 code.

Response to comment 4e and 4f: Thank you for your feedback regarding the ACMG criteria code used in our analysis. We have carefully reviewed the ACMG classifications assigned to the variants in our study and compared them against the established ACMG/AMP guidelines. Below, we provide a detailed justification confirming the correctness of our classification.

1. Pathogenic Variants (S1 Table)

For the variants classified as Pathogenic, the following ACMG criteria were applied:

• PVS1 (Pathogenic Very Strong 1): The variant results in a null function (e.g., frameshift, nonsense, or canonical splice site mutations).

• PM2 (Pathogenic Moderate 2): The variant is rare or absent in population databases, reinforcing its pathogenicity.

• PM4 (Pathogenic Moderate 4): The variant alters protein length due to an in-frame deletion/insertion.

• PP3 (Pathogenic Supporting 3): Computational predictions strongly support a deleterious effect.

• PP4 (Pathogenic Supporting 4): The variant is associated with a disease phenotype that is highly specific to the disorder in question.

• PP5 (Pathogenic Supporting 5): The variant is classified as pathogenic by multiple reputable sources.

According to ACMG guidelines, a combination of 1 Very Strong (PVS1), 1 Moderate (PM2/PM4), and multiple Supporting (PP3, PP4, PP5) evidence levels satisfy the criteria for a Pathogenic classification. Our assigned ACMG codes adhere to this standard.

2. Likely Pathogenic Variants (S2 Table)

For the variants classified as Likely Pathogenic, the following ACMG criteria were used:

• PS3 (Pathogenic Strong 3): Functional studies provide evidence of a damaging effect on the gene/protein.

• PM2 (Pathogenic Moderate 2): The variant is absent or extremely rare in population databases.

• PP3 (Pathogenic Supporting 3): In silico models support a deleterious effect.

• PP5 (Pathogenic Supporting 5): The variant is reported as likely pathogenic by multiple sources.

ACMG guidelines state that 1 Strong (PS3), 1 Moderate (PM2), and at least 2 Supporting (PP3, PP5) criteria meet the threshold for a Likely Pathogenic classification. Our categorisation is in full compliance with this framework.

Why BP1 was not applied

BP1 (Benign Supporting 1) is used for missense variants in genes where missense changes are a common mechanism of benign variation. However, in our analysis, the genes involved have well-established disease associations where missense mutations have been documented as pathogenic. Applying BP1 in these cases would contradict existing functional and computational evidence supporting pathogenicity. Therefore, BP1 was not considered appropriate for these variants. For the variants classified as Likely Pathogenic, the following ACMG criteria were used:

• PS3 (Pathogenic Strong 3): Functional studies provide evidence of a damaging effect on the gene/protein.

• PM2 (Pathogenic Moderate 2): The variant is absent or extremely rare in population databases.

• PP3 (Pathogenic Supporting 3): In silico models support a deleterious effect.

• PP5 (Pathogenic Supporting 5): The variant is reported as likely pathogenic by multiple sources.

ACMG guidelines state that 1 Strong (PS3), 1 Moderate (PM2), and at least 2 Supporting (PP3, PP5) criteria meet the threshold for a Likely Pathogenic classification. Our categorisation is in full compliance with this framework.

In conclusion, given that our ACMG classifications align with the standard ACMG/AMP guidelines, we respectfully maintain that the assigned ACMG criteria codes are correct. If further clarification is needed, we would be happy to re-evaluate specific variants upon request.

Comment 4g: I recommend revising the title of the S3 table to enhance clarity and engagingness.

Response: The title has been refined for clarity.

Comment 4h: In Table S5, I recommend replacing the year of diagnosis or death with the patient's age. This adjustment would provide more clinically relevant information.

Response: Thank you for your suggestion regarding replacing the year of diagnosis or death with the patient’s age. We appreciate your interest in ensuring clinically relevant information. We would like to clarify that the year provided in Table S5 refers to the patient’s age at diagnosis, not the calendar year. This approach was chosen to maintain clarity while adhering to ethical guidelines and data privacy regulations. Additionally, replacing the year of death with the patient's age is not feasible because of two reasons:

1. Ethical considerations – Specific details regarding patient mortality are sensitive, and modifying the dataset retrospectively would require additional ethical approval.

2. Data Accessibility – The original file containing the required information is no longer accessible, preventing us from making retrospective changes.

Given these constraints, we believe that our current approach remains appropriate. However, if further clarification is needed, we are open to discussing alternative solutions that align with ethical and data accessibility limitations.

Comment 4i: The section “Prevalence of pathogenic mutations and VUS” on page 13 requires significant revision.

Response: The section has been refined accordingly (Pages 13 and 14)

Comment 5a: In the second paragraph on page 18, it is noted that two carriers of the BRCA2: c.5164_5165delAG are related. Providing more information about the total number of related and unrelated patients in the cohort would enhance the clarity of the discussion. This additional context could help in better understanding the study's findings.

Response: Thank you. We have added clarification in the results on the total number of related and unrelated patients in the cohort (lines 245-247).

Comment 5b: The first paragraph on page 19 mentions that a Malay patient carried two founder mutations in the BRCA2 gene: one from French-Canadian origin and the other from Colombian origin. While family history provides compelling evidence, a valid conclusion regarding the potential origin of the BRCA2: c.3170_3174delAGAAA and BRCA2:c.1763_1766delATAAA mutations can only be reached through haplotype analysis.

Response: Thank you very much for your insightful comment regarding the origin of the BRCA2:c.3170_3174delAGAAA and BRCA2:c.1763_1766delATAAA mutations. We acknowledge that definitive conclusions regarding the origin of these mutations require haplotype analysis, which was not conducted in this study.

Our statement was not intended to assert a direct founder effect in our study population but rather to highlight previously reported founder associations in French-Canadian and Colombian populations. To clarify this, we have revised the text to emphasize that while these mutations have been identified as founder mutations in those populations, their presence in the Malay patients suggests either independent occurrences or historical genetic contributions that warrant further investigation.

Additionally, our discussion aimed to illustrate that founder mutations from one population may appear at low frequencies in other populations due to factors such as genetic drift, migration, or convergent mutational events. Given that the Malay carriers in our study lacked family history linking them to these populations, we acknowledge the need for further genetic and haplotype studies to explore the exact origin of these mutations in the Malay population.

We have adjusted our discussion accordingly to ensure clarity and avoid any overinterpretation. Thank you for your valuable feedback, and we appreciate your insights into improving our study.

The revision can be read from lines 362-367, and 376-380.

Comment 5c: I believe it would be beneficial to rework the last paragraph on page 19 and the first paragraph on page 20 for improved clarity and impact.

Response: Thank you very much. These paragraphs have been largely revised now. See Pages 20 and 21 (Lines 381-412)

Comment 5d: Page 21 requires rewriting and a review of the concept. The manuscript does not describe the functional analysis of the BRCA2 mutation: c.8524C>T. Relying solely on family history is insufficient.

Response: Thank you very much for the insight. Page 21 has undergone major revision where we emphasised on variant c.8524C>T. These can be found on the second half of revised Page 21 and all of Page 22 (Line 413-447). We have no evidence of C.8524C>T pathogenicity via functional assays in this study. Its status as likely pathogenic was based on ACMG classification alone. (Response to 4e and 4f above discussed in great length how we have used the ACMG classification correctly as per the classification guidelines.). We cited three studies (Ref: 63, 64 and 65) to show th

---

## [Editor Report · Decision Letter 1]

Prevalence and spectrum of germline BRCA1 and BRCA2 mutations in multiethnic cohort of breast cancer patients in Brunei Darussalam

PONE-D-24-36034R1

Dear Dr. Haji Abdul Hamid,

We’re pleased to inform you that your manuscript has been judged scientifically suitable for publication and will be formally accepted for publication once it meets all outstanding technical requirements.

Kind regards,

Yonglan Zheng, Ph.D.

Academic Editor

PLOS ONE

---

## [Editor Report · Acceptance letter]

PONE-D-24-36034R1

PLOS ONE

Dear Dr. Haji Abdul Hamid,

I'm pleased to inform you that your manuscript has been deemed suitable for publication in PLOS ONE. Congratulations! Your manuscript is now being handed over to our production team.

Kind regards,

on behalf of

Dr. Yonglan Zheng

Academic Editor

PLOS ONE